# Role of Plant Virus Movement Proteins in Suppression of Host RNAi Defense

**DOI:** 10.3390/ijms24109049

**Published:** 2023-05-20

**Authors:** Anastasia K. Atabekova, Anna D. Solovieva, Denis A. Chergintsev, Andrey G. Solovyev, Sergey Y. Morozov

**Affiliations:** 1A. N. Belozersky Institute of Physico-Chemical Biology, Moscow State University, 119992 Moscow, Russia; asya_atabekova@mail.ru (A.K.A.); solovyev@belozersky.msu.ru (A.G.S.); 2Department of Virology, Biological Faculty, Moscow State University, 119234 Moscow, Russia; l_anna2000@mail.ru (A.D.S.); ledumpalustre86@gmail.com (D.A.C.)

**Keywords:** virus infection, plant silencing, small RNA, RNA interfering, antiviral response, viral suppressors of RNA silencing

## Abstract

One of the systems of plant defense against viral infection is RNA silencing, or RNA interference (RNAi), in which small RNAs derived from viral genomic RNAs and/or mRNAs serve as guides to target an Argonaute nuclease (AGO) to virus-specific RNAs. Complementary base pairing between the small interfering RNA incorporated into the AGO-based protein complex and viral RNA results in the target cleavage or translational repression. As a counter-defensive strategy, viruses have evolved to acquire viral silencing suppressors (VSRs) to inhibit the host plant RNAi pathway. Plant virus VSR proteins use multiple mechanisms to inhibit silencing. VSRs are often multifunctional proteins that perform additional functions in the virus infection cycle, particularly, cell-to-cell movement, genome encapsidation, or replication. This paper summarizes the available data on the proteins with dual VSR/movement protein activity used by plant viruses of nine orders to override the protective silencing response and reviews the different molecular mechanisms employed by these proteins to suppress RNAi.

## 1. Introduction

In plant virus infections, once a virus genome enters the initial cell(s), the further systemic infection involves cell-to-cell genome movement via plasmodesmata (PD) connecting neighboring cells in plant tissues and subsequent long-distance movement via the vascular system [1,2,3,4,5,6,7,8]. In general, the processes of virus spread from the initially infected cell throughout the plant, as well as viral genome transfer between host organisms, are subject to the major selective forces in the evolution of viruses. As a result, viruses have evolved combinations of at least several genes whose products interact with cellular components to ensure the production of viral genome progeny, the spread of this progeny throughout the organism and, in most cases, the protection of viral genomic nucleic acids to be transferred to other plants. One group of viral genes has been found to be significantly conserved in evolution. In general, these most conserved genes encode the proteins whose activity is directly involved in the replication of the viral genomes [9]. Other viral genes are less conserved, they were even found to be completely unrelated to non-replicative genes in other groups of plant viruses. In particular, these genes encode evolutionarily from very early evolved capsid proteins (CP), which, together, with replication proteins, constitute the so-called “Virus Hallmark Proteins” [9], movement proteins (MP), which apparently first evolved in streptophyta [3,4,7,8,10], viral suppressors (VSR) of post-transcriptional (PTGS), and transcriptional (TGS) gene silencing [11,12,13,14,15,16,17]. In general, to neutralize the effect of RNAi, eukaryotic viruses have evolved to acquire VSRs that significantly reduce the effect of plant antiviral defense. These proteins are critical for successful viral infection, and many viruses contain more than one gene that blocks different steps of RNA silencing [12,13,14]. Importantly, VSR proteins have evolved both in plant- and animal-infecting viruses as well as in non-viral microbial pathogens [12,13,14,15,16,17,18].

MPs, in addition to VSRs, are considered to be a specialized virus-specific system for the colonization of land plants having a symplastic organization in which the PD channels serve as cytoplasmic connections between adjacent cells [19,20,21,22,23]. It can be hypothesized that at least some plant viral MPs have originated from VSRs of ancient ancestral viruses. In general, VSRs, which are important effectors of virus spread in the plant, show remarkable sequence diversity among viruses, and many cases of plant viral proteins performing both functions (MP and VSR) have been revealed (Table 1) [10,11,12,13,14,15,16,17,18,19]. In this review, we consider the structure and activities of such viral proteins with the dual function of MP/VSR.

## 2. Order Tymovirales

### 2.1. Family Alphaflexiviridae

*Potato virus X* (PVX) is a member of the genus *Potexvirus* (family *Alphaflexiviridae*, order *Tymovirales*) [24]. The monopartite RNA genome of PVX encodes a replicase protein with a major domain of RNA-dependent RNA polymerase (RdRP), a movement module called “triple gene block” (TGB) consisting of three genes (TGB1, TGB2 and TGB3) and a capsid protein [25,26,27]. TGB1-encoded protein represents an SF1 RNA helicase and performs dual functions in cell-to-cell movement and RNA silencing suppression. In principle, the influence of TGB1 on silencing suppression could result from inhibition of the silencing signal in the initially infected cells, or, alternatively, from prevention of transfer of the signal to neighboring uninfected cells. Experimental analysis of these possibilities indicates that the effect of PVX TGB1 probably depends on the intracellular stage of local silencing initiation [28]. Further analysis has demonstrated that certain protein mutants are movement deficient but fully competent as VSRs [29]. In general, it can be concluded that the TGB1 protein sub-domains involved in the suppression of silencing are not required for the cell-to-cell trafficking of PVX through PD and the formation of the movement-competent form of the virus genome [30,31]. Importantly, the PVX TGB1 VSR activity accounts for the protein function as a major determinant of viral pathogenicity [32,33] and may influence host transcription [34].

TGB1 proteins of other potexviruses also function as VSRs (Table 1). Indeed, TGB1 proteins of *Plantago asiatica mosaic virus* (PlAMV), *Asparagus virus 3* (AV3), *White clover mosaic virus* (WClMV), and *Tulip virus X* (TVX) have been shown to exhibit a significant, although variable, ability to suppress RNA silencing of a transgene [35]. Interestingly, TGB1 of PlAMV, WClMV, and TVX show a much stronger VSR activity than that of PVX, while the AV3 TGB1 appeared to be a weaker suppressor [35]. To explain the results observed, the authors have hypothesized that the TGB1 of potexviruses, even with significant sequence similarity, have evolved different mechanisms to counteract host antiviral RNA silencing [35]. Taking into account that many VSRs have been identified as ‘pathogenicity determinants’ or ‘symptom determinants’ [32,33], it can also be speculated that the level of suppressor activity is correlated with potexviral infectivity in *N. benthamiana*. However, although TVX TGBp1 is shown to be a strong VSR, TVX fails to infect *N. benthamiana.* By contrast, AV3 consistently infects *N. benthamiana* [35]. These observations demonstrate that the TGB1 RNA silencing suppressor activity does not determine the mode of viral infection in *N. benthamiana*. Interestingly, the nucleolus-localized mutant TGB1 proteins of *Alternanthera mosaic virus* (AltMV) and PlAMV have been found to exhibit significantly higher VSR activity than wild type proteins [36,37]. These TGB1 mutants are competent for cell-to-cell movement and may even give rise to a more efficient artificial bipartite potexviral vectors for whole-plant infection [37].

The fine molecular mechanisms of VSR activity in TGB1 proteins have been first studied for PVX (Figure 1). PVX TGB1 has been shown to interact with AGO1, AGO2, AGO3, and AGO4 and somehow causes the degradation of AGO proteins [38]. The amount of AGO1 accumulated in the presence of PVX TGB1 appears to be dramatically lower than that in mock-infiltrated probes but can be restored when treated with a proteasome inhibitor MG132. This result suggests that the counter-defense role of PVX TGB1 is based on the degradation of AGO proteins via the proteasome pathway. Further support for this idea comes from the observation that plants treated with MG132 are less susceptible to both PVX and *Bamboo mosaic virus*, another potexvirus. Thus, the process of TGB1-AGO interaction inhibits the formation of virus-specific RISC complex (RNA-induced silencing complex) and protects viral genomic RNA from degradation [38,39]. Alternatively, TGB1 of PlAMV interacts with SGS3 and RDR6 and localizes to SGS3/RDR6 bodies to inhibit the amplification of RNA silencing, namely, the formation of secondary virus-specific siRNAs (Figure 1) [39]. If PlAMV TGB1, in addition to its interaction with SGS3/RDR6, has an AGO-affecting activity such as PVX TGB1, the stronger VSR activity of the PlAMV protein compared to that of PVX (see above) can be explained by the use of a dual anti-silencing mechanism [40].

### 2.2. Family Betaflexiviridae

This family of filamentous plant viruses (order *Tymovirales*) includes two sub-families, namely, *Quinvirinae* with three genera and *Trivirinae* with 10 genera [24,41,42]. The genus *Carlavirus* belongs to the family *Quinvirinae*, and its representative filamentous *Potato virus M* possesses a monopartite genome with an organization similar to that of PVX except for an additional cysteine-rich protein (CRP) gene in the 3′-terminal region of the genomic RNA [41]. PVM TGB1 has the VSR activity as PVX TGB1 [43]. PVM TGBp1 differs from PVX TGBp1, which can suppress both intracellular silencing and silencing signal spread, by suppressing only the spread of silencing signal between cells (Table 1 and Figure 1) [43]. In addition, the PVM CRP gene encodes the second VSR of this virus. The CRP VSR has been found to suppress both intracellular silencing and the intercellular spread of silencing, thus apparently increasing the reliability and stability of the carlavirus counter-defense system. Indeed, although PVM TGB1 does not affect viral accumulation at the protoplast level and does not complement the full function of PVX TGB1, it can block the short-range movement of the GFP silencing signal. Moreover, its effect on systemic silencing is found to be rather modest, showing that this VSR might act as a suppressor specifically at the cell-to-cell silencing signal movement step. These data are also supported by the observations made in experiments with TGB1 and CRP expression from a PVX vector: in inoculated leaves, both proteins are found to induce a comparable increase in the PVX RNA levels, whereas, in systemically infected leaves, the effect of PVM TGBp1 is much more modest compared to that of CRP [43].

The viruses belonging to the sub-family *Trivirinae* do not encode TGB and instead contain a single MP gene following the replicase gene [42]. Remarkably, this MP, like TGB1, also possesses the VSR activity. *Apple chlorotic leaf spot virus* (ACLSV) belongs to the genus *Trichovirus* and encodes the P50 protein, which represents the class of tubule-forming MPs [7,44,45]. This protein has also been shown to perform VSR activity. This protein cannot suppress local silencing in leaves infiltrated with an expression vector. However, systemic silencing in upper leaves induced by both single- and double-stranded RNA can be suppressed by P50, even when expression vectors for RNAi induction and P50 expression are infiltrated in different parts of the leaf (Figure 1) [46,47,48]. These observations suggest that, in addition to being a VSR, P50 is capable of cell-to-cell movement.

Another member of *Trivirinae* is the *Citrus leaf blotch virus* (CLBV, genus *Citrivirus*). Notably, CLBV 40K MP has VSR activity similar to ACLSV MP. 40K MP is found to suppress intracellular RNA silencing induced by the expression of either single- or double-stranded RNA, but not cell-to-cell or long-distance spread of the silencing signal (Table 1 and Figure 1) [49]. Long-distance suppression has been tested using two approaches: (i) simultaneous agroinfiltration of the GFP silencing inducer and the suppressor in the same leaf area of *N. benthamiana* 16c plants transgenic for the GFP gene and (ii) concurrent agroinfiltration of both constructs at the tip and the base of single leaves [49]. Therefore, the suppression mechanism of the CLBV MP seems to be different from that of its ortholog in ACLSV, as the latter protein interferes with the systemic spread of the silencing signal without affecting intracellular silencing (see above). This unexpected result can be explained by the rather low sequence similarity of these MP/VSR proteins (24% identity), although both proteins are encoded by viruses belonging to the same subfamily [24].

### 2.3. Family Tymoviridae

Unlike members of the *Alphaflexiviridae* and *Betaflexiviridae* families, viruses in the *Tymoviridae* family have icosahedral virions. *Turnip yellow mosaic virus* (TYMV) is the best-studied member of this family. Unusually, the TYMV P69 MP gene is located in the 5′-proximal region of the genomic RNA and overlaps significantly with a replicase gene [50]. TYMV P69 has been shown to be essential for virus spread and affects symptom severity in infected plants but is dispensable for virus replication in single cells [51,52,53]. In MP proteins of viruses in the *Tymovirus* genus, the N-terminal region is more conserved than the C-terminal part and contains several conserved sequence motifs including blocks of positively charged amino acids [54].

TYMV P69 represents a virulence factor that inhibits dsRNA production and silencing amplification. Interestingly, the P69 expression has also been shown to be associated with increased accumulation of several plant miRNAs [55,56]. Consistent with these observations, TYMV infection suppressed ssRNA-induced silencing but not inverted repeat-induced silencing. Furthermore, the accumulation of endogenous ta-siRNAs, which depend in their production on RDR6, is inhibited in plants expressing the TYMV P69, whereas the accumulation of endogenous siRNAs, which is dependent on DCL2 but not RDR6, is not affected in plants expressing the TYMV P69 VSR. Moreover, P69 has been shown to localize to siRNA bodies where RDR6 resides and where silencing signal amplification occurs, suggesting that P69 inhibits the amplification step that occurs in siRNAbodies, thereby reducing the production of secondary siRNAs [56]. Therefore, P69 can be added to the list of VSRs that inhibit silencing amplification (Table 1 and Figure 1), which includes, in particular, the PlAMV-encoded VSR protein TGB1, which interacts with SGS3 and RDR6 and localizes to SGS3/RDR6 bodies (see above, Section 2.1).

Interestingly, TYMV has evolved an additional mechanism to further suppress RNA silencing. Viral infection induces the expression of an RNaseIII-like cellular enzyme RTL1, which cleaves perfect dsRNA duplexes of endogenous or exogenous origin. A similar mechanism of enhancing RNA silencing suppression is also found for the *Tobacco rattle virus* (family *Virgaviridae*) [56].

## 3. Order Martellivirales

### 3.1. Genus Tobamovirus

The type species of the genus *Tobamovirus* (family *Virgaviridae*) is the *Tobacco mosaic virus* (TMV), the first virus discovered. The TMV genome contains the 5′-proximal open reading frame (ORF) encoding the P126 protein, a component of the viral replication complex; a read-through of the P126 weak terminator codon results in the synthesis of P183, the viral replicase [57]. The 5′-distal region of the TMV genome encodes the 30-kDa MP, the CP, and a small ORF-X product found in a subset of tobamoviruses [57,58]. The functions of TMV MP have been extensively studied, and the current model of the TMV transport mechanism postulates a tight link between virus replication and cell-to-cell movement [59]. According to this model, the P126 and P183 proteins interact with ER membranes to form viral replication compartments (VRCs) which are targeted to PD by viral MP; as a result, the synthesis of progeny virus RNA occurs at the PD entrance, therefore, facilitating MP-mediated transport of virus genomic RNA through PD [59]. Some observations suggest that VRCs can be transported through PD into neighboring cells [60].

The TMV P126 protein has been shown to be a VSR capable of suppressing newly forming silencing response, but not preventing accumulation of siRNA or reversing previously established silencing (Table 1 and Figure 1) [61,62]. Similar activity has been reported for P126 of *Odontoglossum ringspot virus*, another tobamovirus [63]. Further studies have shown that P126 can bind siRNA duplexes and miRNA biogenesis intermediates that also represent small RNA duplexes, thereby preventing siRNA and miRNA incorporation into RISC [64,65]. In addition, P126 has been found to inhibit 3′-methylation of small RNAs that is attributed to binding to siRNA/miRNA duplexes [65]. Interestingly, three TMV P126 regions, namely the methyltransferase domain, the helicase domain, and a portion of a non-conserved region between these domains, have been shown to exhibit the VSR function [66], suggesting that P126 may have more than one mechanism for silencing suppression (Figure 1). Thus, tobamoviruses have evolved to suppress the plant silencing defense response by the 126K protein, which, not being a *bona fide* MP, is directly involved in cell-to-cell movement-coupled virus replication.

A recent analysis of *Tomato mosaic virus* strain N5 (ToMV-N5), which causes systemic necrosis in certain tomato cultivars, has identified MP as a determinant of virus pathogenicity [67]. Compared with a non-necrogenic ToMV strain, ToMV-N5 has three amino acid substitutions in the MP sequence, one of which is responsible for the ability to induce systemic necrosis. Furthermore, the mutations in ToMV-N5 MP result in increased virus accumulation without affecting the efficiency of virus movement from cell to cell. Moreover, the mutant MP has gained the ability to suppress silencing [67]. ToMV-N5 MP has no effect on siRNA accumulation, suggesting that it may function as a VSR at a step downstream of siRNA production [67]. At present, it is not clear how subtle changes in the MP sequence can lead to the acquisition of the VSR function. Other experiments suggest that TMV MP may actually promote rather than suppress RNA silencing. Indeed, TMV MP is unable to affect the levels of GFP mRNA, siRNA, or miRNA in the patch agroinfiltration assay, thus showing no VSR activity, but, on the contrary, enhances the local and systemic spread of silencing [68]. The latter effect is not directly related to the virus cell-to-cell movement function of the protein, as dysfunctional and mislocalized MP mutants are still able to enhance the silencing movement. The role of this MP property in virus infection has been analyzed in experiments with GFP-expressing TMV typically producing fluorescent infection foci in which GFP fluorescence in the center of the foci is inhibited by antiviral silencing. This silencing effect is only observed when TMV MP is expressed from the TMV genome and not found under conditions of MP expression from transgenic plants [68]. These data indicate that TMV MP expressed from the viral genome may contribute to the initiation and/or maintenance of silencing in infected cells as well as to the short-range and systemic transport of silencing. Conceivably, the promotion of antiviral silencing by the virus-encoded protein can be considered as a means of viral self-attenuation, preventing excessive virus accumulation detrimental to the host plant [68]. Furthermore, it would be interesting to know whether the MP of the non-necrogenic ToMV strain (see above), similar to the TMV MP, is able to promote the spread of silencing. Such ability would imply that point mutations in this MP could convert the silencing-promoting protein into a VSR. Taken together, the data available for tobamoviruses, which encode proteins that suppress and promote the silencing defense response, emphasize that compatible virus–host interaction in a given virus–host pair requires a tightly regulated balance between plant defense and virus counter-defense.

### 3.2. Genus Tobravirus

Viruses of the genus *Tobravirus* have bipartite genomes. RNA1 encodes two components of the viral replicase complex, with the larger protein component being expressed by read-through translation of the stop codon of the smaller replicase subunit, the viral MP and a cysteine-rich VSR protein, whereas RNA2 encodes the viral CP and two or more proteins involved in the virus transmission by nematodes [69]. The 16-kDa VSR (16K) of *Tobacco rattle virus* (TRV), the type tobravirus, has been shown to suppress GFP silencing induced by single-stranded and double-stranded RNA in agrobacteria infiltration assays, to reduce the amount of GFP-specific siRNA and to cause a delay in the local and systemic spread of silencing signals [70,71,72]. In the context of TRV infection, 16K is required for the virus to transiently invade the meristem [70]. The mechanism of silencing inhibition by 16K is not fully understood. 16K is unable to bind siRNA and inhibit the activity of existing siRNA but prevents the formation of siRNA-guided RISC complexes, consistent with the observed direct interaction between 16K and AGO4 [73]. 16K has been shown to localize to both the cytoplasm and the nucleus [72]. However, it is not clear whether the nuclear localization of 16K is related to its VSR function. Indeed, in the nucleus, 16K is found to interact with coilin and relocalize this Cajal bodies-resident protein to the nucleolus that is accompanied by activation of the salicylic acid-mediated defense response and restriction of systemic TRV infection [74].

TRV RNA1-encoded MP has also been shown to suppress silencing under certain conditions (Table 1). Indeed, TRV RNA1 deficient in 16K expression retains the ability for systemic transport in plants and silencing suppression. The latter function is found to be dependent on TRV MP [75]. On the other hand, the VSR function of TRV MP is not detectable in patch agroinfiltration assays, suggesting that silencing suppression by TRV MP occurs only in the context of RNA1 replication [75]. This unusual behavior of the viral VSR suggests a novel mechanism of silencing suppression that remains to be discovered.

## 4. Order Tolivirales

### Family Tombusviridae

Viruses of the genus *Luteovirus* have positive-stranded RNA genomes containing six ORFs. Two 5′-proximal ORFs encode components of the viral replicase that are translated from the genomic RNA by a mechanism involving ribosomal frameshifting, ORF3 encodes virus CP; a translational readthrough of ORF3 into ORF5 gives rise to a protein required for the virus aphid transmission; ORF4, located in a different reading frame within ORF3, encodes MP [76]. A recently discovered ORF3a, located upstream of and overlapping the CP-encoding ORF3, encodes for a protein factor required for the virus long-distance transport [77]. In addition, a small 3′-proximal ORF6 encodes a protein termed P6 [76]. The *Barley yellow dwarf virus*-GAV (BYDV-GAV) P6 has been found to suppress local silencing in a GFP patch agroinfiltration assay and to inhibit systemic transport of silencing signals in transgenic plants [78]. As shown in studies of another luteovirus, BYDV-PAV, P6 is unable to suppress local silencing but exhibits moderate suppression of systemic silencing; such a difference between the BYDV-GAV and BYDV-PAV P6 proteins has been attributed to the high divergence of their amino acid sequences [79]. Furthermore, the BYDV-PAV ORF4-encoded MP is found to be a weak suppressor of intracellular silencing and a strong suppressor of both local and systemic transport of silencing signals, and the VSR function of polerovirus MP has been confirmed for a closely related BYDV-PAS [79]. These data show that viruses of the *Luteovirus* genus encode two proteins with VSR activity, MP, and P6 but that their activities vary in different viruses of the genus (Table 1 and Figure 1). Interestingly, the *Potato leafroll virus,* which belongs to the genus *Polerovirus* and has a genome organization similar to that of luteoviruses, lacks the P6 gene. The latter virus instead encodes P0, another dedicated VSR, which exhibits a transient weak silencing suppression activity in the patch agroinfiltration assay without any effect on local and systemic transport of silencing [79].

Members of the genus *Dianthovirus* have a bipartite genome encoding four polypeptides. Two 5′-proximal ORFs in RNA1 encode for replicase components that are expressed, similar to viruses of the genus *Luteovirus*, via a ribosomal frameshift; a 5′-distal gene in RNA1 encodes the viral CP, while RNA2 encodes the viral MP [80]. For the *Red clover necrotic mosaic virus* (RCNMV), the best studied dianthovirus, viral infection has been shown to be associated with silencing suppression accompanied by reduced levels of siRNAs and miRNAs; however, individual RCNMV proteins are unable to suppress silencing in a patch agroinfiltration assay [81]. Further studies have demonstrated that suppression requires a combination of viral RNAs and replication proteins, indicating that the VSR function is exerted by the functional viral replication complex rather than its individual components [81]. These observations are consistent with the observed replication-dependent suppression of silencing by TRV MP (see above), potentially suggesting a common mechanism for linking replication and suppression. Although RCNMV MP is not involved in the replication-coupled suppression of silencing detectable by the use of a patch agroinfiltration assay [81], the VSR function for RCNMV MP has been demonstrated in another assay based on the use of a modified *Turnip crinkle virus* (TCV) genome, in which the gene for CP, the TCV VSR, has been replaced by a GFP gene. Such a TCV-GFP reporter construct is restricted to single initially infected cells unless co-expressed with a VSR, which allows TCV-GFP to be transported from cell to cell, resulting in readily detectable fluorescent infection foci [82,83]. Co-expression of TCV-GFP with RCNMV MP is found to result in the complementation of viral transport, thus demonstrating the VSR function of RCNMV MP. Importantly, according to the site-directed mutagenesis data, the cell-to-cell transport function and the VSR function of RCNMV MP can be uncoupled, showing, therefore, that the latter represents a separate MP feature [82,83]. Thus, in the context of viral infection, the RCNMV counter-defense strategy involves silencing suppression mechanisms that are coupled to virus replication and uncoupled from movement.

## 5. Order Sobelivirales

### Genus Sobemovirus

The single-stranded RNA genomes of viruses of the genus *Sobemovirus* have five ORFs [84,85]. ORF3 is translated from a subgenomic RNA and encodes the virus CP. ORF1, ORF2a, and ORF2b are translated from the genomic RNA; the translation of ORF2a depends on a leaky scanning mechanism, while ORF2b is translated as a result of a translational frameshift during the ORF2a expression [84,86]. An additional ORF translated from the sobemovirus genomic RNA is a recently discovered ORFx that overlaps the 5′-portion of ORF2a [85]. The ORF1-encoded protein, termed P1, is dispensable for viral RNA replication in protoplasts and is required for both local and systemic infection of plants [87,88,89], suggesting its involvement in virus cell-to-cell movement. At present, it remains unclear whether the P1-mediated virus cell-to-cell transport involves a transient PD modification or the formation of MP-formed PD-penetrating tubules that serve as conduits for intercellular transport of virions, as has been shown for many other spherical plant viruses [90].

The *Rice yellow mottle virus* (RYMV) P1 protein is found to inhibit post-transcriptional transgene silencing when expressed either from a PVX-based viral vector in *N. benthamiana* [91] or by means of biolistic gene delivery in rice, the natural host of the virus [92]. The P1 proteins of both RYMV and *Cocksfoot mottle virus* (CfMV) can suppress silencing in a GFP patch agroinfiltration assay, as manifested by increased GFP levels and decreased levels of GFP-specific 24-nucleotides-long small RNAs (Table 1 and Figure 1) [93,94]. The P1 proteins encoded by two RYMV isolates have been shown to enhance both the cell-to-cell spread of local GFP gene silencing and the systemic transport of the silencing signal [95]. Interestingly, P1 from a third RYMV isolate blocks the systemic spread of the silencing signal without affecting its local transport [93,94,96], whereas CfMV P1 suppresses both cell-to-cell and systemic silencing movement [94]. These differences can be attributed to the sequence variability of the RYMV and CfMV P1 proteins [92,97]. On the other hand, these data show that the P1 effects on the silencing movement can be uncoupled from the local silencing suppression exhibited by all sobemovirus P1 proteins tested.

The mechanism of silencing suppression by sobemoviral MPs is only partially understood. CfMV P1 binds ssRNA in a sequence-nonspecific manner [98]. However, it does not show binding to siRNA duplexes or longer dsRNA [94], suggesting, therefore, a mechanism different from that of the *Tomato bushy stunt virus* p19 which binds and sequesters siRNA duplexes [99,100], and that of the *Turnip crinkle virus* (TCV) P38 and the *Pothos latent virus* P14 which both bind long dsRNA molecules [101,102].

Expression of RYMV P1 in transgenic rice has been shown to result in developmental abnormalities phenocopying those known for a *dcl4* mutant [95]. This effect can be attributed to the suppressed accumulation of DCL4-dependent TAS3 siRNAs, which play a critical role in establishing organ polarity, and concominant upregulation of ARF transcription factors targeted by TAS3 siRNA [95]. Thus, the sobemoviral P1 protein can suppress the functions of DCL4, which is known as the primary antiviral Dicer in plants, as has also been found for p38, a silencing suppressor of TCV [103]. This P1 function may apparently be responsible for the local silencing suppression but is unlikely to account for the P1 effects on short-range and systemic spread of silencing signals, which can be uncoupled from the local suppression (see above). On the other hand, DCL4 is found to inhibit the TCV exit from the phloem, thereby blocking the systemic movement of the virus. This DCL4 effect is suppressed by TCV p38 [103], suggesting a similar mechanism for sobemoviral P1.

Recently, the tertiary structure of the RYMV P1 protein has been resolved by X-ray crystallography and nuclear magnetic resonance spectroscopy, providing insight into the structural features of the protein relevant to its functions [104]. The RYMV P1 protein is found to consist of two domains, each containing a zinc finger. The two zinc fingers appear to have different affinities for zinc ions and different functions. Indeed, mutagenesis studies show that substitutions of Cys residues in the N-terminal zinc finger abolish both virus movement and silencing suppression functions of the protein [92], whereas mutations of the C-terminal zinc finger drastically affect the protein conformation [104]. Two other identified regions essential for the protein functions include a region enriched in aromatic residues in the N-terminal domain and a stretch of conserved residues at the C-terminus of the protein. Regulation of the P1 function is proposed to involve (i) the protein dimerization through its N-terminal domain, (ii) possible changes in relative spatial positions of two domains due to a flexible α-helical hinge between them, and (iii) sensing the plant redox status due to the sensitivity of the C-terminal zinc finger to oxidants [104]. It should be noted that the exact functions of the discovered P1 structural elements in silencing suppression and virus movement remain unknown. Furthermore, it remains to be elucidated whether these two P1 functions are interrelated.

## 6. Order Picornavirales

### Genus Fabavirus

Viruses of the genus *Fabavirus* (family *Secoviridae*) have bipartite genomes; RNA1 encodes a polyprotein that is autocatalytically processed into proteins required for virus replication, whereas RNA2 encodes two CPs and a viral MP [105]. The latter protein is synthesized in two forms, which are initiated at alternative AUG codons and share a common C-terminus [105]. A shorter MP, termed VP37, has been shown to be a determinant of pathogenicity and host specificity of *Broad bean wilt virus 1* (BBWV1) [106]. In addition, BBWV1 VP37 is found to exhibit VSR activity in both the patch agroinfiltration assay, in which VP37 can increase the GFP fluorescence and decrease siRNA levels, and the TCV-GFP movement complementation test (Table 1) [106]. However, further details on the mechanism of BBWV1 VP37 silencing suppression are currently unresolved.

## 7. Order Patatavirales

### Family Potyviridae

Members of the family *Potyviridae* have a single-stranded RNA genome of positive polarity (approximately 10,000 nucleotides in length) with a 5′-terminus covalently linked to the protein VPg and a polyadenylated 3′-terminus [107]. The genomic RNA contains a long ORF encoding a polyprotein precursor that is proteolytically processed by three viral proteases (P1, Hc-Pro, and Pro) into ten mature proteins: P1, Hc-Pro, P3, 6K1, CI, 6K2, VPg, Pro, NIb, and CP [107,108]. In addition, RNA polymerase slippage during replication results in the generation of a subpopulation (1–2%) of genomic RNA encoding a small frameshift ORF that gives rise to an additional protein, P3N-PIPO, in which the N-terminal portion of P3 is fused to the PIPO polypeptide [109,110]. According to available data, only the P3N-PIPO protein, which is essential for intercellular movement, can be considered a dedicated MP. Other proteins involved in intercellular movements, such as HC-Pro, CP, CI (an RNA helicase), VPg, and 6K2 (a membrane-embedded protein), have various important roles in the viral infection cycle in addition to their movement functions [107,109].

In infected cells, the CI protein forms pinwheel cytoplasmic inclusions that are detected at the PD entrance, can cross the cell wall through PD to form connections between two CI-inclusions in adjacent cells, and are associated with clusters of virions [111,112]. Similar structures are formed at PD entrances when the CI protein is co-expressed with P3N-PIPO. However, when CI is expressed alone, it aggregates in the cytoplasm. Functional CP is essential for potyvirus cell-to-cell and systemic movement, and mutations in key CP-charged amino acid residues that disrupt virion assembly block virus movement [113,114,115]. Moreover, CP has been shown to be detected in and around PD in the form of fibrils that resemble filamentous potyviral virions [111,112]. These data support the idea that potyviruses are transported in the form of virions or other complexes of CP and genomic RNA, or viral ribonucleoproteins (vRNPs). However, there is increasing evidence that potyviruses can be transported from cell to cell in the form of 6K2-formed replication vesicles [116,117]. In both models of potyviral intercellular transport, P3N-PIPO localizes in PD and interacts with CI, targeting it to PD, where CI forms characteristic PD-associated inclusions. The difference between the two models lies in the mode of genome transport through PD. In the first case, the transport form is a virion or vRNP carrying the CI protein at one end, which interacts with P3N-PIPO and/or the CI protein within inclusions at PD. In the second case, the transport form is VRCs formed by the 6K2 protein interacting with CI inclusions at the PD entrances [6].

The VPg and HC-Pro proteins, which interact with each other and with the virion-bound CI protein, are known to form a morphologically distinctive structure at one end of the virions [118,119] that appears to render the virions competent for cell-to-cell transport. In addition, HC-Pro stabilizes CP, which affects virion output and increases progeny infectivity [120,121]. HC-Pro has been the first identified suppressor of RNA silencing [122,123,124]. HC-Pro is a papain-like cysteine protease that interacts with almost all other potyviral proteins and many host factors [107,120]. The N-terminal domain is responsible for virus transmission by aphids and its FWKG-αH1 element is important for protein self-interaction and plays a role in its RNA silencing suppression function [125]. The central domain is responsible for RNA binding and is involved in genome amplification and transport [126,127,128]. It also contains the conserved FRNK box required for RNA silencing suppressor activity [129,130]. The C-terminal domain is a cysteine protease that catalyzes the autoproteolytic cleavage from the polyprotein at a GG dipeptide at the HC-Pro C-terminus [131].

HC-Pro has RNA silencing suppression activity only in viruses of the genera *Potyvirus* and *Rymovirus*, whereas in other genera of the family *Potyviridae,* this function depends on the P1 protein (Table 1) [132,133,134]. For example, in the *Sweet potato mild mottle virus* (genus *Ipomovirus*), P1 has been shown to suppress RNA silencing by binding to AGO1 through GW/WG-motifs, inhibiting both existing and de novo formed AGO1-containing RISC complexes [132]. In addition, there is substantial evidence that the P1 protein enhances the suppressor activity of HC-Pro through an unknown mechanism [135]. Interestingly, to enhance the suppressor activity of HC-Pro, P1 must be derived from the P1/HC-Pro precursor protein rather than simply ectopically expressed [135,136]. For this reason, studies of HC-Pro-induced RNA silencing suppression often involve a fused P1/HC-Pro protein that can be cleaved in vivo by the P1 protease to form two separate proteins, P1 and HC-Pro [136,137]. Since the RNA silencing suppression activity of the virus appears to be important in determining infectivity in potential hosts, the influence of P1 on HC-Pro may be important in the evolution of potyviruses [137]. Although the RNA silencing suppression activity of HC-Pro was discovered years ago, the exact molecular mechanisms have not been fully elucidated. It is currently believed that HC-Pro suppresses DCL2- and DCL4-mediated silencing at different levels [120,138,139,140]. On the one hand, there is a wealth of data suggesting that HC-Pro-mediated silencing suppression is associated with preventing the loading of viral siRNAs into RNA-induced silencing complexes (RISCs) through direct binding and sequestration of small RNA duplexes [130,139,141]. In addition, other studies suggest alternative pathways for HC-Pro-mediated suppression of RNA silencing. Nevertheless, most HC-Pro proteins are thought to suppress RNA silencing by competing for small RNAs. Efficient binding of small RNAs has been demonstrated for HC-Pro encoded by different viruses [120,128,130,139,141,142,143,144]. Some data suggest that HC-Pro inhibits the systemic transport of the RNA silencing signal that moves ahead of viral infection, likely due to its ability to sequester viral small RNA in infected tissues (Figure 1). The FRNK motif in the central domain of HC-Pro is important for the binding of double-stranded small RNAs, and mutations in this motif have a significant effect on siRNA binding. Interestingly, mutations in the FRNK motif that abolish small RNA binding do not significantly affect the silencing suppression activity of the protein in the case of the *Zucchini yellow mosaic* virus (ZYMV) but result in the loss of this ability in the case of *Tobacco etch virus* (TEV) and *Potato virus Y* (PVY) [130]. This contradiction may be due to differences in the affinity of the mutant protein for different populations of small RNAs [145]. More likely, however, this fact suggests that some HC-Pro proteins may have additional mechanisms of RNA silencing suppression, independent of small RNA sequestering. Indeed, there is evidence for another mechanism of RNA silencing suppression by the HC-Pro protein, namely through inhibition of siRNA and miRNA methylation [146,147,148], leading to polyuridylation and degradation of these RNAs [149]. HC-Pro has been shown to prevent methylation of the 3′-terminus of viral siRNA by inhibiting the formation of the methyl group through interference with two key enzymes of the methionine cycle such as S-adenosyl-L-methionine synthase and S-adenosyl-L-homocysteine hydrolase [150]. Alternatively, the inhibition of methylation may be due to direct interaction with the methyltransferase HEN1 and inhibition of its activity (Figure 1). The *Turnip mosaic virus* (TuMV) HC-Pro has been shown to inhibit the methylation activity of HEN1 through physical interaction via the FRNK motif in which the arginine residue plays a key role [129]. Another possible mechanism for the suppression of RNA silencing by HC-Pro is interference with the antiviral function of AGO1. The *Tobacco etch virus* (TEV) HC-Pro has been reported to interfere with the self-regulation of the plant RNA-silencing pathway and increase the expression of miR168 which negatively regulates AGO1 expression by targeting AGO1 mRNA for degradation [151]. In addition, the *Potato virus A* (PVA) HC-Pro has been shown to interact directly with AGO1 in ribosomal complexes. It has been suggested that this interaction may be involved in the repression of potyviral translation [150]. However, according to other data, the direct interaction with AGO1 does not appear to be related to RNA silencing suppression but does affect PVA and TuMV accumulation. HC-Pro has been shown to recruit AGO1 through its WG-motif to facilitate the interaction of AGO1 with CP. This interaction is required for stable encapsidation and systemic spread of infection, as mutations introduced into this motif dramatically reduced virion accumulation but did not affect RNA silencing suppression [152]. On the other hand, experiments with transgenic *Arabidopsis* plants expressing the P1/HC-Pro precursors of TuMV, ZYMV, and TEV have shown that TuMV P1/HC-Pro triggers post-translational degradation of AGO1, but this effect is not observed in plants expressing P1/HC-Pro ZYMV and P1/HC-Pro TEV [136]. Notably, only TuMV HC-Pro, but not ZYMV or TEV HC-Pro, has been shown to specifically inhibit the HEN1 activity, resulting in the accumulation of unmethylated small RNAs that are unable to load into AGO1. Therefore, it is possible that TuMV HC-Pro enhances the autophagic degradation of AGO1 due to its ability to inhibit HEN1 [129]. In view of the variety of available data on silencing suppression by TuMV HC-Pro (small RNA binding via the FRNK motif, inhibition of HEN1 methylation activity by direct interaction with HEN1, and induction of posttranslational degradation of AGO1), it seems likely that TuMV has acquired different mechanisms for silencing suppression (Figure 1).

Transgenic plants constitutively expressing HC-Pro have been shown to exhibit developmental defects associated with the disruption of miRNA function [153,154]. Apparently, the activity of HC-Pro in suppressing RNA silencing causes many of the disease symptoms observed during potyviral infection. For example, it is believed that most of the developmental defects induced by TuMV in *Arabidopsis thaliana* are caused by the interference of HC-Pro TuMV with pathways that depend on the negative regulation involving miRNA. The Arg residue in the FRNK motif of HC-Pro, which is responsible for binding small RNA duplexes, is also responsible for suppressing the host miRNA-mediated response. It is thought that HC-Pro interferes with the assembly or activity of RISC complexes, resulting in increased accumulation of those mRNAs that are otherwise negatively regulated by miRNA-directed cleavage [130,140,145,153].

The function of HC-Pro as a VSR appears to intersect with plant innate immunity. TEV HC-Pro has been shown to interfere with the expression of jasmonate-regulated transcripts [155]. In addition, a mutation in the TuMV HC-Pro FRNK motif that leads to a reduction in the strength of the suppressor results in a decrease in virus titer and a stronger salicylic acid (SA)-mediated innate immune response by [140]. Subsequently, HC-Pro has been shown to act as a negative regulator of SA-binding protein 3, which is necessary to limit virus spread and SA accumulation. Thus, HC-Pro is able to negatively regulate the SA pathway, indicating its critical role in modulating innate protective responses [156]. Other roles of HC-Pro in plant innate defense responses are also known, but it is unclear to what extent they are related to the suppression of RNA silencing. For example, the *Papaya ringspot virus* (PRSV) HC-Pro interacts with papaya calreticulin and appears to modulate calcium signaling-mediated defense response [157].

As described above, the mechanisms by which HC-Pro can suppress RNA silencing are highly variable. Given the limited number of studies comparing the function of HC-Pro in RNA silencing suppression and its transport function, it is difficult to extrapolate these data. An apparent correlation between the involvement of TEV HC-Pro in long-distance transport and its VSR function has been demonstrated: mutant viruses defective in long-distance transport have no suppressor activity while retaining the ability to replicate and move from cell to cell, albeit less efficiently [126,127]. Similarly, a recent study has shown that a mutant TuMV defective in HC-Pro silencing suppressor activity is incapable of systemic infection but recovers the long-distance movement when DCL2/DCL4 or rdr1/rdr6 are knocked out [139]. Thus, based on the available data, it can be concluded that the involvement of HC-Pro in long-distance transport correlates with its VSR function, whereas RNA silencing suppression and the role of HC-Pro in intercellular transport are distinct functions.

Recently, it has been shown that HC-Pro induces the formation of cytoplasmic potyvirus-induced RNA granules (PGs) during PVA and TuMV infections, which contain plant proteins such as acidic ribosomal protein P0, AGO1, oligouridylate-binding protein 1 (UBP1), varicose (VCS), and eukaryotic initiation factor iso4E (eIF(iso)4E). PGs are associated with the RNA silencing suppression activity of HC-Pro, as point mutations that disrupt this activity of HC-Pro also abolish its ability to generate PGs [158]. Of particular interest are the recently emerging data on the influence of VPg on these PGs. Specifically, a mutation that disrupt the interaction between VPg and eIF(iso)4E has been shown to reduce the transport of viral RNA into the HC-Pro-induced PGs and to impair the ability of VPg to suppress RNA silencing [121,159]. It is possible that, due to the disrupted VPg-eIF(iso)4E interaction, RNA is directed into PGs with less efficiency and is, therefore, more susceptible to degradation via RNA silencing [121]. The interaction of VPg with HC-Pro in PG is also necessary for its stability, as PG-associated HC-Pro can undergo NBR1-mediated degradation if not protected by VPg [160]. VPg has been shown to increase the stability of viral RNA within PGs by a mechanism that is entirely dependent on HC-Pro and other components of PG [158,161]. Thus, the coordinated action of HC-Pro and VPg helps to protect the viral RNA from silencing: HC-Pro forms PG for RNA protection, while VPg prevents its autophagic degradation and promotes increased translation of viral RNA. In addition to the important role of VPg in the function of silencing-associated PG bodies, it has been shown that VPg itself has a weak VSR activity (Figure 1) [162]. In particular, VPg has been shown to affect only RNA silencing pathways induced by ssRNA, but not by dsRNA [162,163]. These data suggest that VPg targets a stage of RNA silencing that is necessary for the generation of dsRNA by plant RDR6. Indeed, VPg of various potyviruses has been shown to interact with plant SGS3 [164,165]. Interestingly, although VPg is predominantly nuclear localized, the VPg-SGS3 interaction occurs in small cytoplasmic bodies that are not 6K2-induced vesicles or PGs; these structures also contain RDR6 and are likely to be previously described sites of dsRNA synthesis [164,165,166]. TuMV VPg has been shown to interact with SGS3 in these bodies, leading to degradation of SGS3 its protein partners such as RDR6. Furthermore, VPg-mediated degradation of SGS3 and RDR6 occurs via both the 20S proteasome and autophagy [165]. In addition, deletion analysis shows that VPg interacts with both the N-terminal domain of SGS3 and its XS domain, which is responsible for RNA binding [167]. This potentially may indicate VPg-mediated inhibition of SGS3 binding to target RNA [165].

## 8. Order Mononegavirales

Plant rhabdoviruses are enveloped bacilliform viruses with negative-sense ssRNA genomes capable of replication in both host plants and specific arthropod vectors [168]. The viral genomes (unsegmented in the genera *Cytorhabdovirus* and *Nucleorhabdovirus* and bipartite in the genera *Dichorhavirus* and *Varicosavirus*) have five canonical genes encoding nucleocapsid protein (N), phosphoprotein (P), matrix protein (M), glycoprotein (G), and polymerase (L) [168,169].

In the citrus strain of *Orchid fleck virus* (OFV-citrus) (genus *Dichorhavirus*), the silencing suppressor function of the virus MP has been found. This non-structural protein is encoded by a highly conserved gene in RNA-1. The OFV-citrus MP has a diffuse cytoplasmic localization and is additionally localized to PD. Importantly, MP associates with structural N and P proteins and directs these proteins from the nucleus to the cell cytoplasm, and subsequently the complex of MP and N (nucleocapsid) proteins is transported to PD [170,171]. The MP is found to be incapable of local silencing suppression of a GFP transgene in a patch agroinfiltration assay or in the TCV movement complementation assay. However, the OFV-citrus MP inhibits the systemic, but not cell-to-cell, transport of silencing signals (Table 1 and Figure 1) [170,171,172]. The mechanism of silencing suppression by the OFV-citrus MP remains to be elucidated.

In contrast to members of the genus *Dichorhavirus*, the structural (virion) P protein of monopartite plant rhabdoviruses has been shown to have a silencing suppressor function [173,174,175,176]. The P protein interacts with AGO proteins, RDR6 and SGS3; accordingly, the P protein has been shown to inhibit the activity of AGO1-containing RISC complexes and the amplification of RNA silencing by interfering with the synthesis of secondary siRNAs [174,175]. Other rhabdovirus proteins have also been shown to be silencing suppressors. In *Barley yellow striate mosaic virus*, the structural P3 protein has been shown to have silencing suppressor activity in addition to the P protein [176], whereas in the *Rice yellow stunt virus* the only identified suppressor is the virion P6 protein [177].

## 9. Order Reovirales

### 9.1. Spinareoviridae

The *Rice ragged stunt virus* (RRSV) is a member of the genus *Oryzavirus* in the family *Spinareoviridae*. RRSV infects plants of the Graminae family and is transmitted in a persistent manner by the brown plant hopper (*Nilaparvata lugens*). The RRSV genome contains 10 double-stranded RNAs ranging in size from 1.2 to 3.9 kb [178]. The RRSV has an icosahedral virion with flat spikes approximately 65–70 nm in diameter [179].

The RRSV RNA binding Pns6 protein has been reported to complement the cell-to-cell movement of a movement-deficient TMV in *N. tabacum* and *N. benthamiana* plants [180,181]. However, Pns6 is unable to support the long-distance movement of a TMV mutant lacking the TMV CP in *N. benthamiana*. The Pns6-eGFP fusion protein transiently expressed in *N. benthamiana* epidermal cells is mainly localized mainly close to the cell wall as well as in punctate sites corresponding to PD [180]. Expression of Pns6 can enhance the pathogenicity of PVX in *N. benthamiana*. RRSV Pns6 has been shown to suppress silencing in the patch agroinfiltration assay in GFP-transgenic *N. benthamiana* plants [182]. More specifically, RRSV Pns6 suppresses local silencing induced by sense RNA, but has no effect on that induced by dsRNA [182]. These data suggest that Pns6 targets the initial steps of RNA silencing upstream of dsRNA production. This is consistent with the fact that RRSV Pns6 preferentially binds ssRNAs, and deletion of a region involved in RNA binding abolishes the silencing suppressor activity of Pns6 (Table 1 and Figure 1) [178,181,182]. Another RRSV protein has also been identified as a VSR. RRSV Pns9 has been found to restore GFP fluorescence upon agroinfiltration of *N. benthamiana* with a 35S-GFP construct. Both Pns6 and Pns9 proteins inhibit local silencing, but only Pns6 affects the long-range movement of the silencing signal [178].

### 9.2. Sidoreoviridae

The *Rice dwarf virus* (RDV) and *Rice gall dwarf virus* (RGDV) are members of the genus *Phytoreovirus* in the family *Sedoreoviridae*. These viruses have icosahedral double-shelled virions with an average diameter of approximately 70 nm [183,184]. Phytoreoviruses are transmitted to rice plants by insect vectors (*Nephotettix cincticeps* or *Resilia dorsalis*) and multiply in both the plants and the invertebrate insect vectors. In RDV, six nonstructural proteins are products of genomic RNA components S4, S6, S10, S11, and S12, respectively [185,186,187,188]. The RDV Pns6 protein has been found to support the cell-to-cell movement of the movement-defective PVX in *N. benthamiana*. This MP has been observed to localize near or within the cell walls of epidermal cells [186]. Strikingly, RDV has evolved to acquire a specialized MP for cell-to-cell movement in a plant vector. When ingested by *N. cincticeps* during insect feeding on infected plants, RDV first infects the filter chamber epithelium. After the assembly of progeny virions, RDV spreads to adjacent organs and eventually moves to the salivary glands from where it can be introduced into rice hosts [189]. Several studies show that RDV can use tubules composed of the viral non-structural protein Pns10 to facilitate the spread of virus particles in the body of *N. cincticeps* [190]. These results indicate that RDV Pns10 may act as a determinant of viral intercellular movement in insects. It has been shown that specific interactions between Pns10 and cytoplasmic actin allow actin-dependent movement of virions between insect vector cells [190]. It is important that in infected cells, the tubules formed by Pns10 interact with the tropomodulin protein, which prevents the dissociation of actin filaments. As a result, the complex of the tropomodulin protein and Pns10 prevents the dissociation of filaments and promotes the movement of tubules along the actin filament [190,191].

Pns10, encoded by genomic segment S10, has also been identified as a VSR [185,192]. The silencing suppression activity of Pns10 has been identified by transient expression in transgenic *N. benthamiana* plants carrying a GFP transgene. The green fluorescence intensity remains strong in the patches co-infiltrated with 35S-GFP and 35S-S10 constructs for a nine day observation period. GFP-specific siRNAs levels are significantly reduced in leaves co-infiltrated with 35S-GFP and 35S-S10 constructs compared to 35S-GFP-infiltrated leaves [185]. Pns10 has been shown to suppress local and systemic ssRNA-induced silencing but not dsRNA-induced silencing (Figure 1) [185].

In grafting experiments with transgenic plants expressing GFP and Pns10, the silencing signal is unable to spread beyond specific veins in systemic leaves. In the case of dsRNA-induced silencing, Pns10 appears to interfere with signal perception in the recipient tissue but does not inhibit silencing signal formation [193]. Interestingly, Pns10 does not directly interact with RDR6 in yeast two-hybrid and co-immunoprecipitation experiments. Therefore, Pns10 likely affects the expression of the RDR6 gene rather than its activity [193]. Among other multiple protein functions, mutagenesis analysis has revealed that Pns10 is able to bind siRNA duplexes with 2-nt 3′ overhangs, and this protein ability is absolutely required for its silencing suppressor activity [193]. Recently, another RDV nonstructural protein, Pns11, has been shown to function as a VSR. Interestingly, while Pns10 is localized to the cytoplasm, Pns11 is localized to both the nucleus and chloroplasts. Pns11 has two bipartite nuclear localization signals (NLSs), and both NLSs are required for Pns11 silencing activity [194].

## 10. Order *Geplafuvirales*, Family *Geminiviridae*

Viruses of the family *Geminiviridae* belong to the recently introduced virus phylum Cressdnaviricota, which includes hundreds of discovered circular Rep-encoding ssDNA viruses [195,196]. Most members of the genus *Begomovirus* have bipartite ssDNA genomes, but other members of this genus and the members of thirteen other genera have monopartite genomes. These genomes typically encode six to seven proteins. In most cases, the virion DNA-sense transcripts contain two partially overlapping ORFs (V2 and V1, encoding CP), whereas the complementary-sense mRNAs usually encode four proteins, namely a replication-associated protein (C1; Rep), which is essential for the rolling-circle replication of the viral genome by the plant DNA polymerases alpha and delta: a transcription activator protein (C2; TrAP), a replication enhancer protein (C3; REn), and a rather small C4 protein (85–100 amino acids long). The gene for the latter protein is located within the C1 gene [197,198,199,200,201,202].

It has long been recognized that the V2 protein in the viruses of the genus *Mastrevirus* performs functions of a membrane-bound MP and works in coordination with CP to enable the trafficking of virus ssDNA from cell to cell through PD [203,204,205]. Evidence for V2 MP activity of is currently emerging for a number of other monopartite geminiviruses [206,207,208,209,210]. Interestingly, the V2 protein has been shown to interact directly with the C4 protein, dramatically enhancing the accumulation of this viral protein [211,212], which, in turn, is capable of participating in virus cell-to-cell movement [213,214,215]. Consistent with this, at least a fraction of C4 molecules may be concentrated in PD [216,217].

In general, many studies have shown that C1 and C2 proteins encoded by different geminiviruses can suppress silencing (Table 1 and Figure 1). Being pathogenicity determinants of geminiviruses, both V2 and C4 proteins also exhibit VSR activity [215,218,219,220,221,222,223,224,225,226,227,228]. Detailed molecular mechanisms of silencing suppression by these MP/VSR proteins in silencing suppression have been elucidated. In particular, the V2 protein of the *Tomato yellow leaf curl virus* (TYLCV), and other members of the genus *Begomovirus* is a strong suppressor of gene silencing (PTGS) that acts as an interactor of SGS3 and suppresses secondary siRNA production [226,227,229,230,231,232,233]. In members of the genus *Curtovirus*, V2 also functions as a strong PTGS suppressor by interfering with the RDR6/SGS3 pathway of secondary siRNA formation [226,227].

The C4 protein is known to suppress both PTGS and TGS. To counteract TGS, C4 suppresses SAM synthetase enzyme activity and interacts with AGO4 to eliminate viral genome methylation [210,214,215]. In addition, some C4 proteins specifically interfere with the cell-to-cell and/or systemic spread of silencing [217,223,227,229]. Particularly, to counteract the intercellular spread of the RNA silencing signal, the TYLCV-encoded C4 protein interacts with BARELY ANY MERISTEM (BAM) 1 and 2 proteins, which are plasma membrane-localized receptor-like kinases [216].

## 11. Conclusions: VSRs and New Directions in Plant Protection

In the last decades, RNA silencing-based strategies have been developed for plant protection against viruses. Currently, the main schemes of this strategy include the construction of transgenic plants expressing virus-specific hairpin transcripts and artificial virus-related miRNAs, or the exogenous application of dsRNAs. Host-induced gene silencing (HIGS) is observed in transgenic plants that are genetically engineered to synthesize pathogen gene-targeting sRNAs or dsRNAs that are further processed into siRNAs. These siRNAs are then led to silencing virulence-related genes. Importantly, HIGS is effective against a wide range of plant pathogens and pests, including viruses, viroids, fungi, insects, and nematodes infesting many important crops [234,235,236]. Some biosafety issues are associated with the use of transgenic plants expressing dsRNAs because transcriptional gene silencing by chromatin modification can lead to heritable changes that have adverse effects. This raises public concerns about the safety of genetically modified organisms. In addition, regulatory problems can increase the cost and time required to bring a transgenic crop to market [235]. One of the new alternative approaches, called spray-induced gene silencing (SIGS), allows exogenous interfering dsRNAs to be directly uptaken directly by pest cells or transferred first to plant cells and then to pathogen cells [234,235,236,237,238,239]. It is important to note that locally sprayed RNAs inhibit pathogen virulence in distal untreated leaves because these dsRNAs, or shorter products of their processing, are capable of systemic spread in plants. In addition, such as HIGS, RNAs developed for SIGS can be designed to target multiple pathogens simultaneously. Another key advantage of RNA-based approaches is that RNAs degrade rapidly, unlike traditional fungicides, which can leave harmful residues in ecosystems, [234,235,236,237,238,239]. However, the use of dsRNA sprays in open fields is complicated by the variable efficiency of dsRNA delivery and the rather low stability of the dsRNA on and in the plants. Therefore, specially designed delivery systems have been especially used with the goal to improve the efficacy of dsRNAs. In particular, recent developments in nanoparticle-mediated carriers for dsRNA delivery include layered double hydroxide, carbon dots, carbon nanotubes, gold nanoparticles, chitosan nanoparticles, silica nanoparticles, liposomes, and cell-penetrating peptides [240].

Another newly developed method is CRISPR/Cas system, which shows great promise as a simple and scalable platform that can be efficiently exploited to achieve virus resistance in plants [241,242,243,244,245]. The ability of the multiplex RNA-based CRISPR/Cas system to target both the DNA and RNA and to generate deleterious mutations in a transgene-dependent and transgene-independent manner is a major advantage of this approach. CRISPR/Cas13 systems can target specific endogenous RNAs, viral RNAs, and RNA intermediates of DNA viruses in plants, thus increasing the possibilities for their application in agriculture [242,243,244,245].

It seems reasonable that the promising targets of antiviral RNAs might be key functional regions of viral genomes, especially those encoding MPs and VSRs [246,247]. Indeed, transgenic plants, expressing artificial miRNAs (amiRNAs) to silence viral genes coding encoding dual MP/VSR proteins (Table 1) of the *Turnip yellow mosaic virus* (P69) [247] and potex/potyviruses (TGB1 and HC-Pro) [246], showed significant resistance to these viruses.

Furthermore, studies on engineered defense against animal viruses have revealed fundamentally new methods not previously used in plants. In particular, it has been shown that the inhibition of an enterovirus VSR by a specific artificial peptide that blocks the VSR dimerization leads to a significant increase in the antiviral siRNA activity in animal cells [248]. Thus, direct targeting and inhibition of plant virus VSR/MP proteins by specific macromolecules could be a novel and powerful method to protect plants from virus infection. Considering this approach, it seems logical to apply the method based on a specific antiviral treatment through the use of RNA aptamers to target and inactivate VSR/MP in the infected cells. RNA aptamers are highly structured oligonucleotides generated by an enrichment process called SELEX (Systematic Evolution of Ligands by Exponential enrichment) to specifically bind molecular targets including proteins. By adopting highly structured 3D conformations, RNA aptamers have been shown to discriminate between different and even very similar proteins [249]. Indeed, RNA aptamer-based inactivation of viral protein activity has been shown to significantly enhance antiviral therapeutics, particularly in HIV and SARS-CoV-2 infections [250,251,252]. Since RNA aptamers are highly structured RNAs, it seems very attractive to use these molecules for exogenous application in plants as it has been applied for artificial interfering dsRNA.

## Figures and Tables

**Figure 1 ijms-24-09049-f001:**
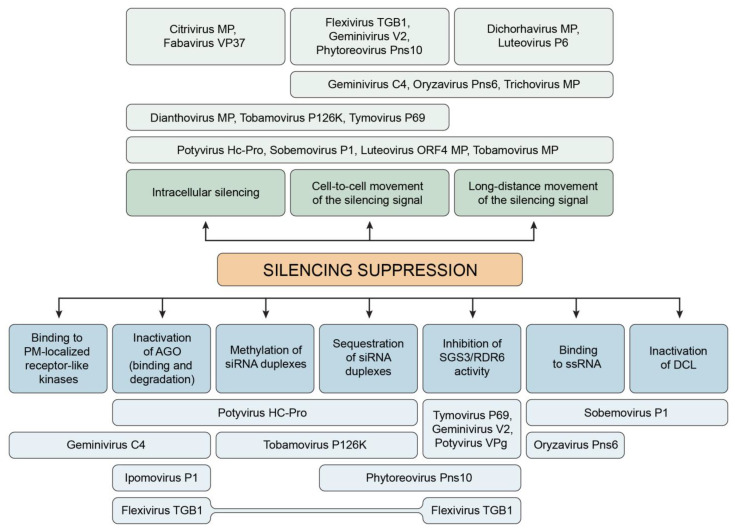
Multifunctional mode of action of VSRs. Specific molecular targets of VSRs in RNA silencing (blue) and intra/intercellular specificities of VSRs (green) are shown.

**Table 1 ijms-24-09049-t001:** A list of the selected plant virus movement proteins acting as RNA silencing suppressors and interfering with host silencing pathway.

Plant Order/Family	Virus Name	MPs
*Tymovirales*/*Alphaflexiviridae*	*Potato virus X*	TGB1
*Tymovirales*/*Alphaflexiviridae*	*Plantago asiatica mosaic virus*	TGB1
*Tymovirales*/*Alphaflexiviridae*	*Alternanthera mosaic virus*	TGB1
*Tymovirales*/*Betaflexiviridae*	*Potato virus M*	TGB1
*Tymovirales*/*Betaflexiviridae*	*Apple chlorotic leaf spot virus*	50K
*Tymovirales*/*Betaflexiviridae*	*Citrus leaf blotch virus*	40K
*Tymovirales*/*Tymoviridae*	*Turnip yellow mosaic virus*	P69
*Martellivirales*/*Virgaviridae*	*Tobacco mosaic virus*	30K
*Martellivirales*/*Virgaviridae*	*Tobacco rattle virus*	RNA2-encoded MP
*Tolivirales*/*Tombusviridae*	*Barley yellow dwarf virus*	ORF4-encoded MP
*Tolivirales*/*Tombusviridae*	*Red clover necrotic mosaic virus*	RNA2-encoded MP
*Sobelivirales*/*Solemoviridae*	*Rice yellow mottle virus*	P1
*Sobelivirales*/*Solemoviridae*	*Cocksfoot mottle virus*	P1
*Picornavirales*/*Secoviridae*	*Broad bean wilt virus 1*	VP37
*Patatavirales*/*Potyviridae*	*Tobacco etch virus*	Hc-Pro
*Patatavirales*/*Potyviridae*	*Potato virus Y*	Hc-Pro
*Patatavirales*/*Potyviridae*	*Sugarcane mosaic virus*	Hc-Pro
*Patatavirales*/*Potyviridae*	*Turnip mosaic virus*	VPg
*Patatavirales/Potyviridae*	*Potato virus Y*	VPg
*Mononegavirales/Rhabdoviridae*	*Orchid fleck virus*	RNA1-encoded MP
*Reovirales/Spinareoviridae*	*Rice ragged stunt virus*	Pns6
*Reovirales/Sidoreoviridae*	*Rice dwarf virus*	Pns10
*Geplafuvirales/Geminiviridae*	*African cassava mosaic virus*	AC4
*Geplafuvirales/Geminiviridae*	*Mungbean yellow mosaic virus*	AC4
*Geplafuvirales/Geminiviridae*	*Tomato leaf curl Palampur virus*	AC4
*Geplafuvirales/Geminiviridae*	*Cotton leaf curl Multan virus*	C4
*Geplafuvirales/Geminiviridae*	*Cotton leaf curl Multan virus*	V2
*Geplafuvirales/Geminiviridae*	*Croton yellow vein mosaic virus*	V2
*Geplafuvirales/Geminiviridae*	*Croton yellow vein mosaic virus*	C4
*Geplafuvirales/Geminiviridae*	*Tomato yellow leaf curl virus*	V2
*Geplafuvirales/Geminiviridae*	*Beet curly top virus*	V2
*Geplafuvirales/Geminiviridae*	*Beet curly top virus*	C4
*Geplafuvirales/Geminiviridae*	*Papaya leaf curl virus*	V2

Note that TGB1 represents TGB1 protein.

## Data Availability

Not applicable.

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
