# Peer review of "Role of Plant Virus Movement Proteins in Suppression of Host RNAi Defense"

_ijms, 2023, doi:10.3390/ijms24109049_

Round 1

Reviewer 1 Report

The review is very informative and well written. Though, the manuscript could be improved by paying attention to the following issues:

Table 1 looks confusing and needs to be organized.

It will be helpful for readers to grasp the whole picture if the authors provide a figure (or figures) summarizing the mechanisms of action of the viral suppressors.

The statement in the Conclusions is very intriguing. It will be nice if the authors give more detailed statements on this topic (RNA silencing-based strategy for plant protection), possibly in a separate part of the manuscript other than the Conclusions.

Typos and mistakes are found in the manuscript. English in some parts of the manuscript looks strange. Thus, the manuscript needs to be carefully read again.

Author Response

The review is very informative and well written. Though, the manuscript could be improved by paying attention to the following issues:

Table 1 looks confusing and needs to be organized.

- Table 1 is re-organized.

It will be helpful for readers to grasp the whole picture if the authors provide a figure (or figures) summarizing the mechanisms of action of the viral suppressors.

- Summarizing figure is added.

The statement in the Conclusions is very intriguing. It will be nice if the authors give more detailed statements on this topic (RNA silencing-based strategy for plant protection), possibly in a separate part of the manuscript other than the Conclusions.
- Conclusions are modified and contain more information on RNA silencing-based strategy for plant protection.

Comments on the Quality of English Language
Typos and mistakes are found in the manuscript. English in some parts of the manuscript looks strange. Thus, the manuscript needs to be carefully read again.
- English Language has been improved throughout the manuscript.

Reviewer 2 Report

The manuscript "Role of plant virus movement proteins in suppression of host RNAi defense" presented by Atabekova AK, et al is clearly written and easy to follow. To further improve the manuscript, it might be helpful if the authors can add some figures in the review. For example, the last paragraph in section 5, the authors could draw a figure to show the structure of RYMV P1 protein, highlight the two zinc fingers and other identified regions essential for the protein functions. Line 470-524, the authors could draw a summary figure for the possible mechanism of RNA silencing suppression by HC-Pro protein. 

Minor issues:

1. Table 1 is not cited in the main text. Please cite.

2. In table 1, TGB1 is gene, not protein, please correct or use more accurate expression.

The English is sufficient to delivery the information clearly. Some minor editing might be needed.

1. Line 13, 'As a counter-defensive strategy, viruses...' should be 'In a counter-defensive strategy, viruses...' since the strategy is from plant not form viruses.

2. Line 652-653, 'six nonstructural proteins are ..., respectively' should remove 'respectively' since the authors didn't point out the specific six nonstructural proteins, there is no corresponding relationship with the RNA components. 

Author Response

To further improve the manuscript, it might be helpful if the authors can add some figures in the review.

For example, the last paragraph in section 5, the authors could draw a figure to show the structure of RYMV P1 protein, highlight the two zinc fingers and other identified regions essential for the protein functions.

-This figure is included into the corresponding paper (see reference 104), and we do not feel that it should be re-drawn in the review.  

Line 470-524, the authors could draw a summary figure for the possible mechanism of RNA silencing suppression by HC-Pro protein.
- Summarizing figure 1 is added.

Minor issues:
1. Table 1 is not cited in the main text. Please cite.

-Table 1 is cited in the main text.

  1. In table 1, TGB1 is gene, not protein, please correct or use more accurate expression.

-In table 1, reference to TGB1 protein is indicated in footnote.

Comments on the Quality of English Language
The English is sufficient to delivery the information clearly. Some minor editing might be needed.
- English Language has been improved throughout the manuscript.

Reviewer 3 Report

Manuscript ijms-2393339 is a thorough and elegant review of plant virus-encoded movement proteins that act as suppressors of RNA silencing. It is a very nice addition to the state-of-the-art on viral silencing suppressors. In addition, the manuscript is very well written. Nonetheless, a few minor editorial changes could be considered for improving its overall

Line 16: Change life to infection

Line 56: Make sure the information is well justified in the four columns of Table 1

Line 62: Change RdRp to RdRP

Line 79 and throughout the manuscript: Virus names should not be italicized or capitalized when referring to a biological entity.

Line 93: Why potexvirus in the name of the virus? Wouldn't virus suffice?

Line 177: The genus Carlavirus belongs ...

Lines 150-151: Italicize N. benthamiana

Line 388: Is 'alpha ' missing before -helical hinge?

Line 604: Change RNA-1 to RNA1

Line 625: Graminae should not be italicized

Line 27: ... movement via the vascular ...

Line 47: Change pathogenes to pathogens

Lines 118-119: ... similar to PVX except for the presence of an additional ...

Line 120: Change as to like

Lines 135-136: ... and instead contain a single MP gene [42]. Remarkably ...

Line 138: ... representing a class of ...

Line 139: ... also VSR activity but cannot suppress local ...

Line 157: Change are to being

Line 160: Eliminate combined

Line 165: In MP proteins of viruses in the genus Tymovirus, the N-terminal ...

Line 277: Viruses of the genus ...

Line 313: Change in parallel to consistent

Line 733: In the last decades, RNA silencing-based strategies have been developed for plant protection against viruses. Currently, ...

Author Response

It is a very nice addition to the state-of-the-art on viral silencing suppressors. In addition, the manuscript is very well written. Nonetheless, a few minor editorial changes could be considered for improving its overall

Line 16: Change life to infection

-It is done.

Line 56: Make sure the information is well justified in the four columns of Table 1

-Table 1 is re-organized.

Line 62: Change RdRp to RdRP

-It is done.

Line 79 and throughout the manuscript: Virus names should not be italicized or capitalized when referring to a biological entity.

-It is done.

Line 93: Why potexvirus in the name of the virus? Wouldn't virus suffice?

-It is changed accordingly.

Line 177: The genus Carlavirus belongs ...

-It is changed accordingly.

Lines 150-151: Italicize N. benthamiana

-It is done.

Line 388: Is 'alpha ' missing before -helical hinge?

-Symbol “α” is added.

Line 604: Change RNA-1 to RNA1

-It is done.

Line 625: Graminae should not be italicized
-It is done.

Comments on the Quality of English Language
Line 27: ... movement via the vascular ...

It is done.

Line 47: Change pathogenes to pathogens

-It is done.

Lines 118-119: ... similar to PVX except for the presence of an additional ...

-It is done.

Line 120: Change as to like

-It is done.

Lines 135-136: ... and instead contain a single MP gene [42]. Remarkably ...

-It is changed.

Line 138: ... representing a class of ...

-It is changed.

Line 139: ... also VSR activity but cannot suppress local ...

-It is changed.

Line 157: Change are to being

-It is done.

Line 160: Eliminate combined

-It is done.

Line 165: In MP proteins of viruses in the genus Tymovirus, the N-terminal ...

-It is changed.

Line 277: Viruses of the genus ...

-It is changed.

Line 313: Change in parallel to consistent

-It is done.

Line 733: In the last decades, RNA silencing-based strategies have been developed for plant protection against viruses. Currently, ...
-It is changed accordingly.